# Digital Form for Assessing Dentists’ Knowledge about Oral Care of People Living with HIV

**DOI:** 10.3390/ijerph19095055

**Published:** 2022-04-21

**Authors:** Ricardo Roberto de Souza Fonseca, Rogério Valois Laurentino, Silvio Augusto Fernandes de Menezes, Aldemir Branco Oliveira-Filho, Ana Cláudia Braga Amoras Alves, Paula Cristina Rodrigues Frade, Luiz Fernando Almeida Machado

**Affiliations:** 1Biology of Infectious and Parasitic Agents Post-Graduate Program, Federal University of Pará, Belem 66075-110, PA, Brazil; ricardofonseca285@gmail.com; 2Virology Laboratory, Institute of Biological Sciences, Federal University of Pará, Belem 66075-110, PA, Brazil; valois@ufpa.br; 3School of Dentistry, University Center of State of Paá, Belem 66075-110, PA, Brazil; menezesperio@gmail.com; 4Study and Research Group on Vulnerable Populations, Institute for Coastal Studies, Federal University of Pará, Braganza 68600-000, PA, Brazil; olivfilho@ufpa.br; 5School of Dentistry, Federal University of Paá, Belem 66075-110, PA, Brazil; ac.amoras@ig.com.br; 6Tropical Medicine Nucleus, Federal University of Pará, Belem 66055-240, PA, Brazil; paulacrfrade@gmail.com

**Keywords:** community dentistry, oral diseases, HIV, public health, oral health

## Abstract

Oral lesions are usually the first sign of HIV infection. The present study aimed to determine the level of the knowledge of dentists on the dental care needs of People Living with HIV (PLWH). This cross-sectional study was conducted between February and May 2021, in the Brazilian state of Pará, during which a total of 51 dentists received an anonymous digital form (Google^®^ Forms Platform) composed of four blocks of discursive, dichotomous, and multiple-choice questions. The questions referred to various aspects of the dental care needs of PLWH, together with data on the professional activities of the dentists. After signing the term of informed consent, the dentists were divided into six subgroups according to the time (in years) since completing their bachelor’s degree in dentistry. The data were presented as descriptive statistics and percentages, and then analyzed using the Kappa test. Most (70.6%; 36 of 51) of the dentists were female, the mean age of the dentists was 32.5 years, and a majority (80.2%) were based in the city of Belem; the mean time since graduation was 8.5 years, with 22 (43.1%) having more than 5 years of professional experience, and 31 (60.8%) having graduated from a private dental college. Just over half (51%) of the 51 dentists had completed graduate courses, and the most common dental specialty was orthodontics (19.6%). Most (74.5%) of the dentists work in the private sector, 38 (74.5%) claimed to have already provided oral care to PLWH, and 43 (84.3%) had access to specialist content on the oral care needs of PLWH. In terms of the knowledge of the dentists with regard to the oral care needs of PLWH, four of the ten diagnostic questions obtained more inadequate answers than expected, whereas the final two questions (11–45.1% and 12–31.4%) demonstrated that many of the dentists adopt unnecessary modifications in their oral care protocol for PLWH, due to a fear of contamination. Overall, our results demonstrate a frequent lack of knowledge, especially with regard to the oral healthcare needs of PLWH, which may account for many of the stigmas that persist in the dental care of this vulnerable group.

## 1. Introduction

In the early 1980s, the first cases of Acquired Immunodeficiency Syndrome (AIDS) were recorded around the world, and innumerable reports drew the attention of the scientific and medical community to the high mortality rates in the patients affected by this new pathogen [1,2]. Initially, this epidemic was associated with specific groups, such as men that have sex with men, drug users, and recipients of blood transfusions [3,4].

A few years after this first report, the pathogen was isolated and then classified as the Human Immunodeficiency Virus (HIV) [5,6,7]. Since this time, a great deal of research has been conducted to understand the characteristics of the virus, including aspects such as the HIV–host interaction, HIV pathogenesis, transmission, diagnosis, and treatment. Infection typically begins with an acute phase of intense replication of the virus, which then spreads to the lymphoid tissue. This is followed by a chronic phase, which is often asymptomatic, and may progress to an advanced and severe phase of immunosuppression, known as AIDS [8,9,10,11,12].

Even today, HIV infection is a major public health concern, in particular, in developing nations such as Brazil, where infection rates vary considerably among the country’s different regions, according to data from the Brazilian Ministry of Health (BMH). The 2019 epidemiological bulletin of the BMH showed that the northern region had the second lowest rate of notification of new HIV infections among Brazil’s five regions, with a total of 4948 (11.8% of all cases). Comparisons with the previous bulletins revealed a tendency for linear growth in the HIV/AIDS detection rate [13,14,15].

It is common for People Living with HIV (PLWH) to present oral manifestations during HIV infection, which may provide systematic criteria for the diagnosis of the acute, chronic, or AIDS phases of the infection, or indicate a possible HIV infection, and also increases the morbidity of PLWH [11,12]. Given this, key groups that are vulnerable to HIV infection require regular oral care, which is essential for the treatment of HIV, considering that the principal focus of oral care should be the prevention, diagnosis, and treatment of all oral lesions potentially associated with HIV. This is why dentists must be prepared to assist PLWH, given that adequate oral care will play a direct role in the improvement of both oral and systemic health, a decrease in morbidity, and the control of HIV infections [16,17].

Dentistry courses in Brazil have an extensive curriculum that aims to prepare dental students for the correct treatment of individuals of all kinds [18]. Despite their extensive training, studies of Brazilian dentists and dental students have demonstrated a major lack of knowledge with regard to the adequate oral care of PLWH. Most of these studies have been conducted in the northeastern, southeastern, and southern regions of Brazil, and have shown that this lack of knowledge leads to a fear of being contaminated by HIV during dental procedures, in particular, surgeries, or by the aerosol produced by dental machinery [19,20,21,22,23].

In addition to a fear of HIV contamination, published studies have also shown that many dentists and dental students are not well acquainted with the lesions associated with HIV, and that many stigmas associated with this virus persist in these professionals, which unfortunately limit their adoption of the biosafety protocols necessary to satisfy the oral care needs of PLWH, due to a lack of knowledge on HIV contamination, transmission, and replication [19,20,21,22,23]. Overall, the available data indicate that public dental health services in Brazil are very limited in terms of the treatment of PLWH, which may contribute to HIV morbidity and mortality, and a decline in the quality of life in the members of this group. In this context, the present study provides a professional profile of a group of dentists from the state of Pará, and evaluates the level of knowledge that they have on the oral care needs of the state’s PLWH.

## 2. Materials and Methods

### 2.1. Study Design and Area Knowledge

The present descriptive and cross-sectional study involved dentists residing in Pará state, northern Brazil, who were surveyed between February and May 2021 (Figure 1). Pará is the second largest state in northern Brazil, with an area of 1,245,870.707 km^2^, and a population of 8,690,745 inhabitants. It has 144 municipalities, and a Gross Domestic Product (GDP) of R$49.5 billion (approximately US$10 billion), and a per capita GDP of R$7707.

The Brazilian Federal Board of Dentistry indicates that approximately 6834 dentists are currently active in Pará, more than half of which (51%; 3507) are based in the state capital, the city of Belem. Belem has an estimated population of approximately 1,500,000 inhabitants, that is, less than one fifth of the total population of the state, which indicates a severe imbalance in the distribution of dentists in Pará, and, in turn, the availability of oral care, in particular, for key groups, such as PLWH.

The DATASUS/Tabnet platform (Ministry Health database) was accessed for information on the number of oral health procedures conducted in Pará, and identified 273,326 records for the period between January and July 2021. Unfortunately, this platform does not provide any information on the distribution of these procedures in the state or the groups attended, so there are no systematic data on the oral care of PLWH.

### 2.2. Ethics

The present study was approved by the Research Ethics Committee of the Institute of Biological Sciences at the Federal University of Pará (UFPA), under protocol number 4.606.188. Informed consent was obtained from all the participants that were included in the analyses in this study.

### 2.3. Data Collection

A non-probabilistic “snowball” sampling technique was used to invite dentists to participate in the present study [24]. Initially, six dentists from Pará were contacted by the authors. These six dentists received detailed information on the objectives of the study, the data collection procedures, and data security, and were asked to sign the term of informed consent before participating in the study. They were then requested to help publicize the study and invite other dental professionals to participate.

The digital form used to collect data was distributed using WhatsApp, Facebook, Instagram (Facebook Inc., Menlo Park, CA, USA), and Telegram (Telegram Messenger LLP, Moscow, Russia). Each dentist that received and filled out the digital form was requested to invite three other dentists to participate. This procedure was repeated a number of times until completing the study sample.

All participants were informed of the nature of the study, its potential risks and benefits, and were required to sign the digital term of informed consent before being given access to the digital form. The inclusion criteria were: (i) an age of at least 18 years; (ii) graduation from a dentistry course; (iii) active registration in the federal and regional councils of dentistry; (iv) based in Pará state; and (v) with access to the internet. Potential participants who did not sign the informed consent form or did not have stable internet access were excluded from the study. The exclusion criteria were: (i) individuals who moved to other states during the study; (ii) being a dental student or undergraduate; (iii) a lack of internet access; and (iv) any other type of impairment for the completion of the digital form.

The dentists were divided into six subgroups according to the time (in years) since completing their bachelor’s degree in dentistry. The groups were formed and organized to categorize the technical level based on the time completing their bachelor’s degree: G1 group: ≤1 year (control group, it was established because, in theory, newly graduated dentists were the control group because they are leaving college with more up-to-date information compared to those who have graduated longer ago); G2 group: 2 years; G3 group: 3 years; G4 group: 4 years; G5 group: 5 years; G6 group: >5 years.

### 2.4. Sample Collection and Digital Form

A digital form was used to collect data in the present study. This form was formatted and administered using the Google^®^ Forms Platform (Google Inc., Mountain View, CA, USA), and was distributed through an electronic link in the apps mentioned above. The digital form was composed of four blocks of questions, and the participants could not advance to the next block without filling all the questions in the current block [25].

Blocks 1 and 2 contained information on the researchers and the objectives, risks, and potential benefits of the study, as well as the consent form [26]. Block 3 included questions on the age and gender of the participant; the municipality in which they are based; time since completion of their bachelor’s degree in dentistry; type of institution (public or private); graduate courses (*sensu stricto* or *sensu lato*); dental specialty; and professional details, such as their workplace, whether they had already provided oral care to PLWH, or whether they had previous access to specialist content on the oral care needs of this group. Block 4 contained 12 questions on the technical–scientific knowledge of the participant in terms of their knowledge on HIV infection, transmission, and diagnosis, and its clinical signs and symptoms. There were also questions on basic human immunology, dental biosafety, and the clinical examination and management of PLWH patients.

All of the questions in blocks 3 and 4 have dichotomous or multiple-choice answers, in addition to a section inviting the participant to provide details of their own knowledge on the question. To determine whether the digital form helped the dentists to increase their knowledge on the oral care needs of PLWH, a 0–10 self-reported scale was included at the beginning and the end of the digital form to allow each dentist to evaluate their own knowledge prior to and after answering the form. For analysis, scores of 0–6 were considered to reflect insufficient knowledge, and scores of 7–10 were accepted as sufficient knowledge.

### 2.5. Statistical Analysis

All the data collected during the present study were entered into an Excel database (Microsoft Corp., Redmond, WA, USA), and then converted to BioEstat format. Statistical parameters (absolute and relative frequencies, mean, median, amplitude, and standard deviation) were used to describe the characteristics of the sample, in terms of the quantitative and qualitative variables investigated using the digital forms.

The level of knowledge of the dentists on the oral care needs of PLWH was assessed using two approaches. The first was to compare the self-reported knowledge scores from the beginning and end of the form, with the total number of adequate answers given to the questions presented in blocks 3 and 4.

The second evaluation compared the self-reported knowledge scores from the beginning of the digital form with those at the end. Between these two moments, all of the dentists answered questions on a range of topics, which would have permitted the emergence of reflections, doubts, and fears, as well as the definition of appropriate resolutions for these various topics related to the oral care needs of PLWH. The effects of this process are reflected in the replicability of the responses of the dentists prior to and following the 10 questions. The difference was evaluated using the Kappa test, which verified the significance of the possible difference between the first and second responses (based on a *p* < 0.05 significance level). All statistical procedures were run in the BioEstat 5.0 program [27,28].

## 3. Results

### 3.1. Sample

A total of 60 dentists participated in the present study, of which, 54 were contacted following the invitation of the original six dentists. However, nine of these dentists were not included in the analyses, due to not satisfying the inclusion criteria. These individuals included five dentists that did not sign the informed term of consent, three who had not yet graduated, and one that did not fill in the digital form. This meant that the sample analyzed here consisted of 51 dentists based in Pará state.

### 3.2. Professional, Demographic, and Epidemiological Characteristics

The personal and professional characteristics of the 51 dentists are shown in Table 1. The sample was composed primarily of female dentists (70.6%), with only 15 male dentists (29.4%). The mean age of the participants was 32.5 years (range: 18–61 years), and most were based in Belem (80.2%) and Ananindeua (9.8%). On average, the participants had completed their graduation in dentistry 8.5 years prior to the study, although the modal group (43.1% of the total) had graduated more than 5 years ago, followed by the recent graduates (37.2%). Most (60.8%) of the dentists had graduated from a private college.

Overall, 26 (51%) of the dentists had taken a graduate course (*sensu stricto* or *sensu lato*), whereas 18 (35.3%) had no graduate qualifications. The most common dental specialty was orthodontics, practiced by 10 of the dentists (19.6%), followed by dental prosthesis (15.7%), and periodontics (13.7%). The most common workplace was in the private sector (74.5%), followed by the public sector (27.5%), whereas seven (13.7%) of the participants were not working at the time of the study due to the COVID-19 pandemic. Most (74.5%) of the dentists had already provided oral care to PLWH, and 84.3% had access to some type of specialist content on the oral care needs of PLWH.

The answers to the technical and scientific questions on HIV-related topics (block 4) are summarized in Table 2 (see also Appendix A). As mentioned above, some of the questions in blocks 3 and 4 had a discursive section in which the participant could include an evaluation of their own knowledge. These questions are summarized here as Doubts, Questions, or Fears (DQF) for analysis. Most (62.7%) of the dentists differentiated HIV and AIDS correctly (question 1), whereas 72.5% identified the infection mechanism correctly (question 2). However, whereas most dentists (78.4%) knew how HIV transmission occurs (question 3), only 17.6% were able to identify the common oral lesions in PLWH (question 4), and only 31.4% were able to identify the common clinical signs and symptoms of HIV infection.

Less than half (47.1%) of the dentists knew which exams are necessary to diagnose HIV, whereas in the DQF space, 14 (58.3%) of the participants that had answered this question correctly raised further questions on HIV test parameters, demonstrating a certain level of doubt. Most (70.6%) of the dentists knew where to refer the patient after a positive HIV diagnosis (question 7), whereas almost half (47.1%) were unsure on the HIV window period. However, only nine (33.3%) of the 27 participants that answered positively described the period correctly in the DQF space.

When asked about changes in Personal Protective Equipment (PPE) or biosafety protocols for the oral care of PLWH (question 9), 62.7% responded negatively, although most (78.1%) of these dentists questioned the effectiveness of the basic PPE protocol in the DQF, especially in the context of contact with blood or aerosol during the treatment of PLWH. Just over half (52.9%) were in doubt with regard to the prescription of drugs for PLWH, and in the DQF space, they inquired into the drugs and how to prescribe them.

In question 11, 54.9% of the dentists responded negatively to any kind of change during non-invasive dental treatment, even though, in the DQF space, 74.5% of all the dentists, that is, 38 participants, questioned the risk of contamination when coming into contact with saliva or aerosol. In question 12, most of the dentists (68.6%) answered negatively to any kind of change during invasive dental treatment, but once again, many of the participants (43.1%) asked which procedures they should adopt in the case of work accidents in the DQF space.

### 3.3. Self-Reported Knowledge

Prior to completing the questionnaire, 32 (62.7%) of the dentists declared they had sufficient knowledge to care for PLWH (Table 3), whereas at the end of the questionnaire (in light of the wrong answers and the informative texts on the topics covered), 38 (74.5%) declared having improved their knowledge. However, when the answers to the questionnaire were evaluated, only 28 (54.9%) had actually improved their knowledge.

It can be verified through the agreement Kappa test that at the beginning of the associated questionnaire after the 10 specific questions, there was a result of 0.0784, inferring a weak replicability by the kappa value and absence of agreement. Assessing the agreement at the end of the questionnaire after the 10 specific questions, a result of 0.1961 suggests, again, poor replicability of the test application; however, there is agreement, denoting improvement in learning/knowledge about the topic addressed.

## 4. Discussion

The present digital study sought to assess the level of the knowledge of dentists with regard to the oral care needs of PLWH in Pará state, northern Brazil. Although the topic is not new—see Menezes et al. [15], Senna et al. [23], and Muniz et al. [29]—this is the first paper to focus specifically on the oral care of PLWH in northern Brazil. According to the Joint United Nations Program on HIV/AIDS (UNAIDS), AIDS is a global public health problem, and in Brazil, approximately 920,000 individuals are living with HIV, which is known to deplete the immune system, resulting in a reduction of the LTCD4^+^ count, leading to manifestations such as oral lesions. A good basic knowledge of HIV is important, not only for healthcare in PLWH, but, in particular, to reduce the stigmas associated with this condition [30,31].

Infection by HIV can be divided into the acute, asymptomatic, chronic symptomatic, and AIDS phases [15]. It is important for dentists to understand the HIV infection cycle because oral lesions may appear in all phases. The acute phase involves an acute infection which occurs 3–6 weeks after contamination and takes 30–60 days for the production of anti-HIV antibodies. This is known as the HIV window period, during which, the virus is highly transmissible, and the clinical diagnosis of HIV is hampered. The duration of the subsequent, asymptomatic phase is highly variable, and though the transmission of HIV may still occur, there is less risk of this than during the acute phase. Even so, the asymptomatic phase is the period when the appearance of the first HIV-related oral lesions is most likely and is when the early signs of HIV infection are most often apparent to dentists.

Senna et al. [23] showed that only 55% of a sample of 140 dentists from the Brazilian state of Minas Gerais were willing to conduct oral care on PLWH and actually knew how to apply this care correctly. These authors demonstrated that stigmas with regard to HIV contamination, a lack of knowledge, and the risk of occupational accidents were the principal sources of concern among the dentists from Minas Gerais. The results of the present study demonstrate a similar pattern to that found by Senna et al. [23], given that a majority of the dentists provided inadequate answers to many of the questions (blocks 3 and 4), reflecting a lack of good basic knowledge, with the DQF showing that some dentists avoid treating PLWH due to a fear of HIV contamination through contact with aerosols or blood.

Despite the existence of standard occupational prevention measures such as PPEs, the lack of a national protocol for the oral healthcare of PLWH may contribute to the stigmas, technical ignorance, and misinformation on HIV infection [23], which may result in discomfort for PLWH during treatment. In a study of 67 PLWH in Pernambuco state, northeastern Brazil, Muniz et al. [29] analyzed the perceptions of PLWH with regard to the approach, reception, and biosafety protocol used by their dentists during oral treatment. In this analysis, 31% of the patients did not inform the dentist about their HIV infection, and of those that did inform the dentist, a third related suffering differential treatment, and 27% reported having suffered discrimination. Most (57%) of the PLWH that hid their HIV status from the dentist did this in order to guarantee proper oral care, and to avoid prejudice. The concerns of these patients are consistent with the findings of the present study, in particular, with regard to the need to change biosafety protocols for both invasive and non-invasive treatments (questions 11 and 12). Lucena et al. [18] evaluated the sociodemographic profile, knowledge, attitudes, and practices of dental students from Rio Grande do Sul state, and found that 90% of the students referred to the need to treat all patients as potentially being infected with HIV, and taking adequate PPE measures due to the fear of HIV contamination, primarily by the aerosols generated during dental procedures. This was similar to the concerns cited in the present study.

Oliveira et al. [21] evaluated the knowledge of dental students and found that most were able to identify oral lesions as Kaposi’s sarcoma, oral candidiasis, and oral hairy leukoplakia, which contrasts with the findings of the present study, given that 82.4% of the dentists had an inadequate knowledge of these oral lesions. Sposto et al. [22] assessed the general and specific knowledge of students before and after attending an informative lecture on HIV, and found that the percentage of correct answers increased from 49% to 54.4%, demonstrating that contact with information improves the knowledge level of the students, which is partly consistent with the results of the present study, given that 74.5% of the dentists declared having improved their knowledge after completing the questionnaire.

The results of the present study indicated that the “veteran” dentists (5 or more years since graduating) had more professional and scientific knowledge than newly graduated dentists (no more than 2 years after graduation). However, the veterans tend to take less care with biosafety procedures [32]. Corrêa et al. [20] found that most dentists, from both public and private workplaces, had insufficient knowledge for the safe treatment of PLWH. Contamination by aerosols produced by rotary dental instruments was a major concern in the present study, and Tellier et al. [33] concluded that the pathogens transmitted by fluids such as blood may remain suspended in the air through the splashes generated by the high-speed rotation of these instruments, although the HIV particles in aerosols may not necessarily cause infections.

Though the present study was successful in terms of the data collected, it did suffer from certain limitations, such as the relatively small sample size, its restriction to the state of Pará, the involuntary exclusion of individuals that did not have access to the internet, a possible bias of helpfulness generated by the invited participants, the answers prepared by the researchers, and, in particular, the identification of the need to assess the satisfaction of PLWH with regard to their dental care needs and treatment at both private and public levels.

## 5. Conclusions

In conclusion, the present study served as the basis for a larger-scale cross-sectional study, and our results demonstrated that the lack of knowledge about oral health care of PLWH will increase stigmas among dentists and prejudice PLWH dental care; therefore, new information and establishment of a national protocol for dental care for PLWH will help dentists to better treat HIV-infected individuals.

## Figures and Tables

**Figure 1 ijerph-19-05055-f001:**
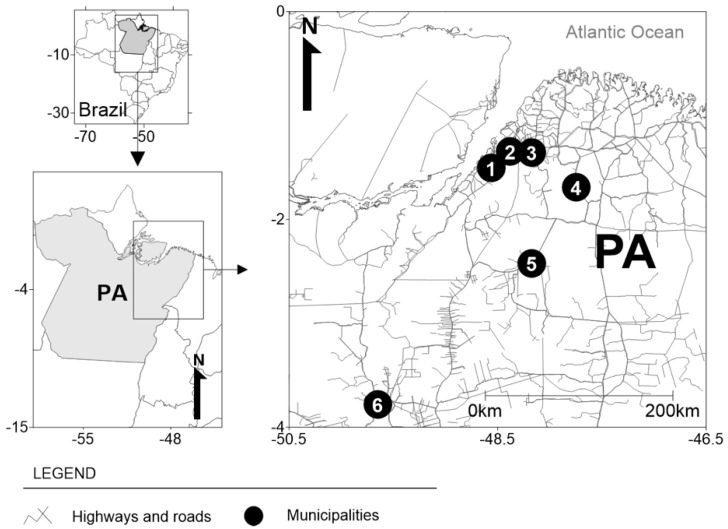
Geographic location of municipalities in the Brazilian state of Pará (PA) where dentists worked and provided information for this study. Cities: (1) Belem, (2) Ananindeua, (3) Benevides, (4) São Domingos do Capim, (5) Tomé Açu, and (6) Tucuruí.

**Table 1 ijerph-19-05055-t001:** Socioeconomic profile and risk behavior related to HIV infection among people with HIV/AIDS.

Variables	Total	CI 95%
Parameters	51	
**Gender ***		
Male	15 (29.4%)	0.17–0.41
Female	36 (70.6%)	0.58–0.83
**Age (years) *****		
18–29	27 (52.9%)	0.39–0.66
30–39	13 (25.5%)	0.13–0.37
40–49	9 (17.6%)	0.07–0.28
50–≥51	2 (4%)	0.02–0.20
**Base (municipality) *****		
Belem	41 (80.2%)	0.69–0.91
Ananindeua	5 (9.8%)	0.01–0.18
São Domingos do Capim	1 (2%)	−0.01–0.05
Benevides	1 (2%)	−0.01 0.05
Tomé Açu	1 (2%)	−0.01 0.05
Tucuruí	2 (4%)	−0.01–0.09
**Time (years) since graduation *****		
≤1 year (control)	19 (37.2%)	0.24–0.50
2 years	5 (9.8%)	0.01–0.18
3 years	3 (6%)	−0.006–0.12
4 years	0 (0%)	null
5 years	2 (3.9%)	−0.01–0.09
>5 years	22 (43.1%)	0.29–0.56
**Type of College ***		
Private	31 (60.8%)	0.47–0.74
Public	20 (39.2%)	0.25–0.52
**Graduate degree ****		
Specialization	26 (51%)	0.37–0.64
Master’s degree	8 (15.7%)	0.05–0.25
Doctorate	1 (2%)	−0.01–0.05
No graduate degree	18 (35.3%)	0.22–0.48
Other type of graduation	6 (11.8%)	0.02–0.20
**Dental specialty ****		
Pediatric dentistry	1 (2%)	−0.01–0.05
Orthodontics	10 (19.6%)	0.08–0.30
Facial and Jaw Orthopedics	1 (2%)	−0.01–0.05
TMJ ^‡^ and Orofacial Pain	0 (0%)	null
Dentistry	4 (7.8%)	0.005–0.15
Geriatric dentistry	0 (0%)	null
Endodontics	6 (11.8%)	0.02–0.20
Periodontics	7 (13.7%)	0.04–0.23
Dental prosthesis	8 (15.7%)	0.05–0.25
Implantology	5 (9.8%)	0.01–0.18
Maxillofacial Prosthesis	0 (0%)	null
Maxillofacial Surgery and Traumatology	2 (3.9%)	−0.01–0.09
Sports Dentistry	0 (0%)	null
Facial Matching	2 (3.9%)	−0.01–0.09
Pathology and Stomatology	1 (2%)	−0.01–0.05
Dental Radiology	0 (0%)	null
Unfinished	13 (25.5%)	0.13–0.37
**Workplace ****		
Private	38 (74.5%)	0.62–0.86
Public	14 (27.5%)	0.15–0.39
Not working currently due to the COVID-19 pandemic †	7 (13.7%)	0.04–0.23
Other type of work in dentistry	2 (3.9%)	−0.01–0.09
**Already provided oral care to PLWH ***		
Yes	38 (74.5%)	0.62–0.86
No	26 (21.6%)	0.37–0.64
Uncertain	2 (3.9%)	−0.01–0.09
**Had access to specialist content on the oral care needs of PLWH ***		
Yes	43 (84.3%)	0.74–0.94
No	7 (13.7%)	0.04–0.23
Uncertain	1 (2%)	−0.01–0.05

* dichotomous inquiry; ** multiple choice inquiry; *** discursive inquiry; ^‡^ TMJ: temporomandibular joint dysfunction; † not working due to COVID-19 reasons.

**Table 2 ijerph-19-05055-t002:** Responses of the participants of the present study to the general and specific questions on the oral needs of people living with HIV.

Question	Answers			
	Adequate n (%)	CI 95%	Inadequate n (%)	CI 95%
Difference between HIV infection and AIDS *	32 (62.7%)	0.49–0.76	19 (37.3%)	0.24–0.50
HIV infection mechanism *	37 (72.5%)	0.60–0.84	14 (27.5%)	0.15–0.39
HIV transmission **	40 (78.4%)	0.67–0.89	11 (21.6%)	0.10–0.32
Common oral lesions in PLWH **	9 (17.6%)	0.07–0.28	42 (82.4%)	0.71–0.92
Common signs and symptoms of HIV infection **	16 (31.4%)	0.18–0.44	35 (68.6%)	0.55–0.81
How to request laboratory tests *	24 (47%)	0.33–0.60	27 (53%)	0.39–0.66
Referral of PLWH to care services *	36 (70.6%)	0.58–0.83	15 (29.4%)	0.16–0.41
The HIV window period *	27 (53%)	0.39–0.66	24 (47%)	0.33–0.60
Biosafety protocol for PLWH dental care *	32 (62.7%)	0.49–0.76	19 (37.3%)	0.24–0.50
Prescription of drugs for dental treatment *	24 (47.1%)	0.33–0.60	27 (52.9%)	0.33–0.60
Changes in the non-invasive care protocol for PLWH *	28 (54.9%)	0.41–0.68	23 (45.1%)	0.31–0.58
Changes in the invasive care protocol for PLWH *	35 (68.6%)	0.55–0.81	16 (31.4%)	0.18–0.44

HIV: human immunodeficiency virus; AIDS: acquired human immunodeficiency syndrome; PLWH: people living with HIV; * dichotomous query; ** multiple choice query.

**Table 3 ijerph-19-05055-t003:** Variation in the knowledge of dentists in the Brazilian state of Pará with regard to the oral care needs of PLWH.

	Knowledge Level				*p*-Value
Moments (Classification)	Sufficient n (%)	CI 95%	Insufficient n (%)	CI 95%	
At the beginning of the form (informed knowledge)	32 (62.7%)	0.49–0.76	19 (37.3%)	0.24–0.50	0.2105 ^‡^
At the end of the form (informed knowledge)	38 (74.5%)	0.62–0.86	13 (25.5%)	0.13–0.37	
After the 10 specific questions (demonstrated knowledge)	28 (54.9%)	0.41–0.68	23 (45.1%)	0.31–0.58	0.0191 ^†^

^‡^ For the difference in self-informed knowledge; ^†^ For the difference observed empirically.

## Data Availability

Not applicable.

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
