# Peer review of "Digital Form for Assessing Dentists’ Knowledge about Oral Care of People Living with HIV"

_ijerph, 2022, doi:10.3390/ijerph19095055_

Round 1

Reviewer 1 Report

Line 23: shouldn’t a full stop be added after 2021? To validate the questionnaire seems like a new sentence.

Line 29: “and with an average of 8.5 years of graduation graduates”. This sentence is not clear, please clarify!

Line 31: the majority of professionals, 26 (51%) completed or completed a stricto sensu postgraduate course. What do you mean by completed or completed? Please clarify! It is also advisable to put the total number also inside the brackets. For example (the majority of professionale (n = 26, 51%))

Line 79: “According to these studies, both dentists 79 and dental students have as their main fear being contaminated by HIV during dental 80 care, especially during invasive procedures or by aerosol, which is common in dental care, 81 other common results of these studies are that both dentists and dental students have dif-82 ficulties in identifying all oral lesions associated with HIV, in addition biosafety measures 83 are frequently exposed as failures of the participants of these studies, occupational acci-84 dents questions and what attitudes to take in face of it are other issue discussed and due 85 to the decreased knowledge of dentists and dental students demonstrated in the literature 86 might be the reason why the difficulties in oral health care and permanence of the stigma 87 in Brazil is still maintained [18-22]”.

This sentence has 6 commas! Please revise and break it down to simple easy-to-understand sentences.

Line 92: Please remove the word “and”.

Materials and methods

Line 105 to 115: Again, the authors should revise the sentence and break it down. As a general guidance: A sentence should have no more than two commas. Please also always use the same number formatting (273.326 or 6,834, comma or dot)

The inclusion criteria indicate that the participant should be working in the state of Para. However, the table shows that some unemployed dentists also participated. Please clarify

It is essential that the actual questions (from which table 2 was generated) are presented in the study (maybe as a supplementary file).

Results

Line 242: The range 18-61 is a bit weird! The authors say that only graduated dentists took part in this survey. Can the authors please clarify whether 18-year-old participants indeed took part in the survey? Were they graduated dentists?

Author Response

Dear Reviewer 1

We would like to thanks all your contributions to the paper, regarding the extensive English editing language it was performed according your specifications to improve the manuscript. also the supplementary material was added as requested

Reviewer 2 Report

General comment: The authors addressed an important and current topic! But when reading, this manuscript has many English writing and grammar errors. The tenses used must be correct, because in several places the authors begin and continue to the present and then to the past and so on.

- Starting with the title, the authors are not very clear with the expression “people living with HIV”. Here, the referee suggests changing the latter to "people infected with HIV" or simply "people with HIV".

- The abstract needs a new reformulation after the correction of the English.

- The introduction is confusing and unclear with the chronological topics described and needs to be reformulated appropriately.

- Line 94: the authors can highlight better their aim here.

-  The authors have used long sentences in several places and this creates confusion to better understand their work, such as lines 105-115, 254-258, 280-287, 288-294 and so on.

- Also, the results section must be checked, because in some places the idea is confused.

- Line 319: the referee suggests to delete “This is a table.”, because it is implied to be a table.

- The discussion section is very long and confusing, here you find the other authors cited and the results of this work are not precisely discussed. It needs to be reformulated and to be clearer with reasonable sentences.

Author Response

We would like to thanks all your contributions to the paper, regarding the extensive English editing language it was performed according your specifications to improve the manuscript. 

Reviewer 3 Report

Relevant in-text citations are missing. Eg., “According to an epidemiological bulletin from the Ministry of Health, the northern region of Brazil has the second lowest rate of notification of HIV infection among Brazilian regions.”

The need for using both “>=18 years” and “graduation in dentistry” as inclusion criteria is unclear. Are there dental graduates less than 17 years of age in this region?

Please provide additional details on how the questionnaire was developed and validated.

Please provide details on how the sample size was calculated.

The text is difficult to follow due to grammatical deficiencies.

- Run-on sentences: eg. Lines 105 to 115.

- Lack of punctuation/capitalization and incomplete sentences: e.g. “and this can drastically impair the quality of life of women. PLWH.”

The manuscript requires major revision for language and grammar. I would suggest the use of a professional English language editing service.

Author Response

We would like to thanks all your contributions to the paper, regarding the extensive English editing language it was performed according your specifications to improve the manuscript. unfortunately the sample size was established by the study answer period

Round 2

Reviewer 1 Report

Thnaks for addressing the comments

Author Response

the manuscript test has been revised and moderate English improvement were made.

Reviewer 2 Report

General comment: The authors have generally edited and improved their manuscript, but there are still things to improve in English and in sentence construction. However, the authors used still long sentences in several places and this creates confusion to better understand their work. They need another revision of the paper!

- In the abstract, in line 21, it is better to start "The aim of the study was to ...".

- In line 57 the verbs "have and use" must be in the past tense, because the sentence doesn't make sense in the present way. In line 84 "It is commonly people ..." is incorrect in English. Lines 88-95, is a huge sentence that needs to be separated and making sense. The same for lines 100-105.

- Lines 196-201: The authors here need to be clearer; please improve the context of the sentence.

- In line 294, also here the sentence needs to be improved, because it is not clear.

- At line 404, "an" is incorrect. The referee suggests deleting the table entered in the discussion, as it creates confusion during the reading.

Author Response

(The authors gave the same response as above.)

Reviewer 3 Report

Thank you for addressing the previous comments.

Although the language is somewhat improved, many grammatical errors still remain. I would suggest that you submit this manuscript for revision by a professional English language editing service.

Author Response

(The authors gave the same response as above.)
